# Cardiovascular Factors Associated with COVID-19 from an International Registry of Primarily Japanese Patients

**DOI:** 10.3390/diagnostics12102350

**Published:** 2022-09-28

**Authors:** Akira Matsumori, Matthew E. Auda, Katelyn A. Bruno, Katie A. Shapiro, Toru Kato, Toshihiro Nakamura, Koji Hasegawa, Ahmed Saleh, Sherif Abdelrazek, Hany Negm, Niyata Hananta Karunawan, Leslie T. Cooper, DeLisa Fairweather

**Affiliations:** 1Clinical Research Center, Kyoto Medical Center, Kyoto 612-8555, Japan; 2Department of Cardiovascular Medicine, Mayo Clinic, 4500 San Pablo Road, Jacksonville, FL 32224, USA; 3Center of Clinical and Translational Science, Mayo Clinic, 4500 San Pablo Road, Jacksonville, FL 32224, USA; 4Department of Immunology, Mayo Clinic, 4500 San Pablo Road, Jacksonville, FL 32224, USA; 5Division of Cardiovascular Medicine, Department of Medicine, University of Florida, Gainesville, FL 32608, USA; 6Department of Clinical Research, National Hospital Organization, Tochigi Medical Center, Utsunomiya 320-8580, Japan; 7Department of Cardiology, National Hospital Organization, Kyushu Medical Center, Fukuoka 810-8563, Japan; 8Division of Translational Research, National Hospital Organization, Kyoto Medical Center, Kyoto 602-8566, Japan; 9Klinikum Braunschweig, Academic Hospital of Hannover Medical School, 30625 Braunschweig, Germany; 10Department of Cardiology and Angiology, University Heart Center Freiburg-Bad Krozingen, 79189 Bad Krozingen, Germany; 11Cardiology and Ultrasonography Unit, Research Institute of Ophthalmology (RIO), Cairo 11261, Egypt; 12Department of Cardiology and Vascular Medicine, Tugurejo General Hospital, Semarang 50185, Central Java, Indonesia; 13Department of Environmental Health and Engineering, Johns Hopkins Bloomberg School of Public Health, 615 N. Wolfe Street, Baltimore, MD 21205, USA

**Keywords:** myocarditis, heart failure, biomarkers, troponin, sex differences

## Abstract

Aims: We developed an international registry to examine cardiovascular complications of COVID-19. Methods: A REDCap form was created in March 2020 at Mayo Clinic in collaboration with the International Society of Cardiomyopathy, Myocarditis and Heart Failure (ISCMF) and data were entered from April 2020 through April 2021. Results: Of the 696 patients in the COVID-19 Registry, 411 (59.2%) were male and 283 (40.8%) were female, with a sex ratio of 1.5:1 male to female. In total, 95.5% of the patients were from Japan. The average age was 52 years with 31.5% being >65 years of age. COVID-19 patients with a history of cardiovascular disease (CVD) had more pre-existing conditions including type II diabetes (*p* < 0.0001), cancer (*p* = 0.0003), obesity (*p* = 0.001), and kidney disease (*p* = 0.001). They also had a greater mortality of 10.1% compared to 1.7% in those without a history of CVD (*p* < 0.0001). The most common cardiovascular conditions in patients with a history of CVD were hypertension (33.7%), stroke (5.7%) and arrhythmias (5.1%). We found that troponin T, troponin I, brain natriuretic peptide (BNP), N-terminal pro-BNP (NT-proBNP), C-reactive protein (CRP), IL-6 and lambda immunoglobulin free light chains (Ig FLC) were elevated above reference levels in patients with COVID-19. Myocarditis is known to occur mainly in adults under the age of 50, and when we examined biomarkers in patients that were ≤50 years of age and had no history of CVD we found that a majority of patients had elevated levels of troponin T (71.4%), IL-6 (59.5%), creatine kinase/CK-MB (57.1%), D-dimer (57.8%), kappa Ig FLC (75.0%), and lambda Ig FLC (71.4%) suggesting myocardial injury and possible myocarditis. Conclusions: We report the first findings to our knowledge of cardiovascular complications from COVID-19 in the first year of the pandemic in a predominantly Japanese population. Mortality was increased by a history of CVD and pre-existing conditions including type II diabetes, cancer, obesity, and kidney disease. Our findings indicate that even in cases where no abnormalities are found in ECG or ultrasound cardiography that myocardial damage may occur, and cardiovascular and inflammatory biomarkers may be useful for the diagnosis.

## 1. Introduction

Late in 2019 an infectious outbreak that caused pneumonia and acute respiratory distress syndrome (ARDS) occurred in Wuhan, Hubei Province, China resulting in the World Health Organization (WHO) naming the virus Severe Acute Respiratory Syndrome Coronavirus-2 (SARS-CoV-2) and the condition Coronavirus disease 2019 (COVID-19) [1,2,3]. By March 2020, the infection had spread around the world and was declared to be a pandemic [1]. Early in the pandemic, a number of international registries were developed to examine cardiovascular outcomes from COVID-19 including the American Heart Association (AHA) COVID-19 Cardiovascular Disease (CVD) registry, the International retrospective Postgraduate Course in Heart Failure registry for patients hospitalized with COVID-19 and CArdioVascular disease (PCHF-COVICAV) registry (15 countries) and the Lean European Open Survey on SARS-CoV-2 (LEOSS) Registry [4,5,6]. In March 2020 we also developed an international registry at Mayo Clinic in collaboration with the International Society of Cardiomyopathy, Myocarditis and Heart Failure (ISCMF) to examine cardiovascular outcomes from COVID-19 with an emphasis on unrecognized myocarditis. Patients in this registry were primarily from Japan, providing the first data on cardiovascular outcomes from COVID-19 in a largely Japanese population.

Myocarditis is well-known to occur more often in males than females and to occur more frequently in younger individuals (<50 years of age) [7,8]. Due to the predominantly younger demographic for myocarditis, it is possible that cases of myocarditis went unreported during the pandemic. It is important to determine the likelihood of myocarditis in COVID-19 patients not only due to acute disease but also because myocarditis can progress to dilated cardiomyopathy (DCM) and chronic heart failure [9,10,11,12]. Saleh et al. reported that a relatively high percentage of patients with COVID-19 with no known history of CVD had elevated cardiac damage and inflammatory biomarkers such as cardiac troponin, N-terminal pro-brain natriuretic peptide (NT-proBNP), and lambda immunoglobulin free light chains (Ig FLC) suggestive of myocarditis [13,14,15]. We developed an international registry at the beginning of the pandemic in March 2020 in collaboration with the International Society of Cardiomyopathy, Myocarditis and Heart Failure (ISCMF) to examine cardiovascular complications of SARS-CoV-2/COVID-19 and whether COVID-19 was associated with myocarditis.

## 2. Materials and Methods

### 2.1. Ethics Statement

Research carried out in this study complied with the Helsinki Declaration. The study was approved by the Mayo Clinic Institutional Review Board and receipt of a waiver of the need to consent subjects was obtained.

### 2.2. International Registry

This international registry included patients hospitalized with COVID-19 based on a positive PCR test for SARS-CoV-2. The registry included several vulnerable populations such as children under the age of 18 years, the elderly greater than 65 years of age, and pregnant women (Table 1). A survey was created in March 2020 using REDCap at Mayo Clinic to identify cardiovascular complications associated with SARS-CoV-2 infection in patients with COVID-19. A link to the REDCap survey was sent to organizations like the World Heart Federation (WHF), the International Society of Cardiomyopathy, Myocarditis and Heart Failure (ISCMF) and the Asian Pacific Society of Cardiology, for example, as well as to some hospitals and/or individual physicians so that providers could access the survey to provide information about their COVID-19 patients for the study. Data were obtained from April 2020 through April 2021 during the initial year of the pandemic.

### 2.3. Statistical Analysis

All statistical analyses were performed using Microsoft Excel and GraphPad Prism. Categorical variables are presented as *n* (%). Continuous data are expressed as mean ± standard deviation. Categorical data are expressed as numbers and percentages and *p*-values were obtained using Fisher’s exact test. Two-group analyses of normally distributed data were performed using Student’s *t* test and the Mann–Whitney *U* test for non-normally distributed data. Proportions were compared using *χ*^2^ test or Fisher’s exact test, where appropriate. A value of *p* < 0.05 was considered significant.

## 3. Results

### 3.1. Patient Characteristics

Patient demographics are shown in Table 1. Of the 696 patients in the COVID-19 Registry, 411 (59.2%) were male and 283 (40.8%) were female with a sex ratio of 1.5:1 male to female. The majority of the patients entered in the registry were from Japan (*n* = 664, 95.5%) with 26 from Germany (3.7%) and 5 from other countries (0.7%), which accounts for primary ethnicities of Asian (*n* = 645, 93.8%), Caucasian (*n* = 21, 3.1%), Hispanic (*n* = 13, 1.9%), Middle Eastern (*n* = 4, 0.6%), and Black (*n* = 2, 0.3%) (Table 1). Out of 696 patients, 219 were >65 years of age (31.5%), while 32 were <18 years of age (4.6%) and 17 (2.4%) were pregnant. The average age of individuals in the registry was 52 years ±21.9, but patients that had a history of cardiovascular disease (CVD) were around 70 years of age while patients without a history of heart disease were much younger at around 45 years of age (no CVD history 44.1 ± 20.0 vs. history of CVD 69.6 ±15.0 years of age, *p* = 6.35 × 10^−58^) (Figure 1). Similarly, patients that required respiratory support or had acute care complications were around 70 years of age while those without these complications were around 45 years of age (Figure 1). CVD in those with a history of heart disease included stroke, coronary artery disease (CAD) (which included myocardial infarct), valvular heart disease, hypertension, arrhythmias, heart failure, myocarditis, cardiomyopathy, and prior coronary artery bypass graft (CABG) surgery.

### 3.2. COVID-19 Symptoms and Outcomes in Registry Patients with or without a History of CVD and by Sex

Symptoms and outcomes of COVID-19 that were assessed as part of the registry included fever, dry cough, fatigue, loss of smell and taste, shortness of breath, sore throat, headache, myalgia, diarrhea, decreased appetite, chills, chest discomfort, nausea, stomach/abdominal pain, vomiting, dizziness/light-headedness, palpitations, angina, confusion/altered mental status, and myocarditis. Dry cough (no CVD history 45.8% vs. CVD history 36.1%, *p* = 0.022) and loss of smell and taste (no CVD history 31.0% vs. CVD history 15.6%, *p* < 0.0001) occurred more often in patients with no history of CVD while shortness of breath (no CVD history 23.6% vs. CVD history 32.7%, *p* = 0.017), decreased appetite (no CVD history 11.3% vs. CVD history 19.0%, *p* = 0.010) and stomach/abdominal pain (no CVD history 1.9% vs. CVD history 4.9%, *p* = 0.043) occurred more often in patients with a history of CVD (Table 2). The only symptom of COVID-19 that displayed a sex difference in the registry was shortness of breath, which occurred more often in males than females (males *n* = 120, 29.6% vs. females *n* = 61, 22.1%, *p* = 0.034).

### 3.3. History of Pre-Existing Lung and Other Conditions in Registry Patients with or without a History of CVD and by Sex

Most patients in the registry did not have a history of pre-existing lung conditions including asthma, COPD, lung cancer, or interstitial lung disease regardless of whether they had a history of CVD (no CVD history 90.3% vs. CVD history 87.0%, *p* = 0.21) (Table 3 and Figure 2a). For those patients with a pre-existing lung condition, there were no differences between the incidence of the condition based on CVD history. However, more COVID-19 patients with a history of CVD had other pre-existing conditions besides those affecting the lung (no CVD history and no other pre-existing conditions 86.2% vs. CVD history and no other pre-existing conditions 68.4%, *p* < 0.0001). COVID-19 patients with a history of CVD had a greater percentage of pre-existing type II diabetes (no CVD history 5.4% vs. CVD history 27.1%, *p* < 0.0001), other types of cancer (no CVD history 3.0% vs. CVD history 10.4%, *p* = 0.0003), obesity (no CVD history 1.7% vs. CVD history 7.3%, *p* = 0.0011), and kidney disease (no CVD history 0.4% vs. CVD history 4.1%, *p* = 0.0012) (Table 3 and Figure 2b). More men than women in the registry had type II diabetes (men 15.2% vs. women 7.7%, *p* = 0.0037) and COPD (men 3.6% vs. women 0%, *p* = 0.0013) while women had more asthma (women 9.3% vs. men 3.8%, *p* = 0.0046) (Table 3 and Figure 2c).

### 3.4. Lung Imaging and Respiratory Support for Registry Patients with or without a History of CVD and by Sex

Around 40–45% of patients in the registry did not receive lung imaging (Table 4). Patients that received imaging had a chest X-ray, CT, or both. In this study, patients with a history of CVD were more likely to be assessed using CT alone (no CVD history 34.8% vs. CVD history 51.0%, *p* = 0.0001). Patients without a history of CVD were more likely to have normal findings on their lung imaging compared to than those with a history of CVD (no CVD history 34.7% vs. CVD history 9.9%, *p* < 0.0001). COVID-19 patients with a history of CVD had more pneumonia (no CVD history 19.4% vs. CVD history 34.7%, *p* = 0.0018) and pleural effusion (no CVD history 1.2% vs. CVD history 5.8%, *p* = 0.017) than those with no history (Table 4). Patients with a history of CVD also received more oxygen (no CVD history 12.3% vs. CVD history 29.3%, *p* < 0.0001) and were more likely to receive mechanical ventilation or be intubated (no CVD history 1.3% vs. CVD history 5.1%, *p* = 0.0094) than those without a history. Women were also more likely to have normal findings on their lung imaging than men (women 36.4% vs. men 18.2%, *p* < 0.0001). Men had more ground glass shadowing (men 51.4% vs. women 37.7%, *p* = 0.009), pneumonia (men 29.0% vs. 19.8%, *p* = 0.042) and bilateral patchy shadowing (men 20.1% vs. women 11.1%, *p* = 0.023) than women.

### 3.5. Mortality and Acute Care Complications in Registry Patients with or without a History of CVD

We found that COVID-19 patients with a history of CVD had a mortality of 10.1% whereas patients without a CVD history had a mortality of only 1.7% (*p* < 0.0001) for index hospitalization (Table 5). Patients with a history of CVD were more likely to be admitted to the ICU (no CVD history 1.1% vs. CVD history 4.0%, *p* = 0.028) and to develop pneumonia following SARS-CoV-2 infection (no CVD history 8.2% vs. CVD history 26.2%, *p* < 0.0001), but time in the ICU and number of days until death did not differ between the two groups. Additionally, those with a history of CVD were more likely to develop acute respiratory distress syndrome (ARDS) (no CVD history 1.1% vs. CVD history 5.4%, *p* = 0.0016) and shock (no CVD history 0.2% vs. CVD history 2.5%, *p* = 0.011) (Table 5). We did not detect a sex difference in mortality in the registry (data not shown).

### 3.6. Cardiovascular Findings in Registry Patients with or without a History of CVD

The most common cardiovascular condition in patients with a history of CVD was hypertension (33.7%) followed by stroke (5.7%) and arrhythmias (5.1%) (Table 6). Only one patient in the registry had myocarditis prior to COVID-19. No COVID-19 patients were diagnosed with myocarditis (Table 2), but 6 patients developed cardiac arrest (Table 5). In patients with COVID-19, heart failure assessed by New York Heart Association (NYHA) class was relatively mild with 3.9% of patients without a history of CVD developing NYHA Class I or II heart failure compared to 11% of those with a history of CVD (*p* = 0.0017) (Table 6). Patients with a history of CVD also had more electrocardiogram (ECG) abnormalities such as arrythmias, atrial fibrillation, ST elevation and AV block (40.6% without ECG abnormalities) than patients without a history of heart disease (81.3% without ECG abnormalities) (*p* = 0.013) (Table 6), although only around 7% of patients in the registry received an ECG.

### 3.7. Cardiovascular and Inflammatory Biomarkers Found in COVID-19 Registry Patients with or without a History of CVD

Cardiovascular and inflammatory sera biomarkers that were above the reference level for patients with or without a history of CVD included troponin T, troponin I, brain natriuretic peptide (BNP), N-terminal pro-BNP (NT-proBNP), C-reactive protein (CRP), IL-6 and lambda immunoglobulin free light chains (Ig FLC) (Table 7 and Figure 3). The only sera biomarkers that were increased in patients with COVID-19 in the registry that had a history of CVD compared to those without a CVD history were peak CRP (*p* = 7.98 × 10^−5^) and kappa Ig FLCs (*p* = 0.0032). Importantly, the remaining sera biomarkers were elevated in COVID-19 patients regardless of whether they had a history of CVD or not resulting in there not being a significant difference between these two groups. This suggests that cardiac damage and inflammation occurred in patients without a history of CVD, which may indicate undiagnosed myocarditis. Cardiovascular symptoms in patients with a history of CVD may be due to past disease and may not represent new-onset issues related directly to COVID-19. Myocarditis is known to occur mainly in adults under the age of 50 [7] and so we examined the percentage of patients with these biomarkers that were ≤50 years of age which were as follows: troponin T (*n* = 5/7, 71.4%), troponin I (*n* = 2/9, 22.2%), BNP (*n* = 1/13, 7.7%), NT-proBNP (*n* = 13/29, 44.8%), CRP (*n* = 99/184, 53.8%), IL-6 (*n* = 50/84, 59.5%), creatine kinase/CK-MB (*n* = 8/14, 57.1%), D-dimer (*n* = 89/154, 57.8%), kappa immunoglobulin FLC (*n* = 3/4, 75.0%), and lambda immunoglobulin FLC (*n* = 5/7, 71.4%). Thus, several biomarkers that have been associated with clinical myocarditis were found to occur in a majority of COVID-19 patients without a history of CVD that are under 50 years of age including troponin T, CRP, IL-6, CK-MB, D-dimer and kappa and lambda immunoglobulin FLCs.

## 4. Discussion

Relatively quickly in the pandemic it became clear that COVID-19 was associated with a number of cardiac complications including arrythmias, acute coronary syndromes, hypercoagulability, thrombosis, and myocarditis [16,17]. We developed an international registry in collaboration with the International Society of Cardiomyopathy, Myocarditis and Heart Failure (ISCMF) to examine cardiovascular complications from SARS-CoV-2/COVID-19 and to determine whether COVID-19 was associated with myocarditis. We found that patients with COVID-19 with a history of CVD, especially hypertension, were significantly older than those without, that they had increased mortality, and that they had elevated biomarkers associated with cardiac damage and heart failure including elevated levels of troponin T, IL-6, creatine kinase/CK-MB, D-dimer, kappa and lambda Ig FLC (see summary of findings in Figure 4). Additionally, a significant portion of young patients under 50 years of age with COVID-19 that did not have a history of CVD had elevated biomarkers suggesting undiagnosed myocarditis.

None of the 696 patients in the registry that had COVID-19 were diagnosed with myocarditis although 6 patients developed cardiac arrest and one patient previously had myocarditis (Table 6). However, we found that COVID-19 patients with a history of CVD had increased mortality (10.1%) compared to those without a CVD history (1.7%) (Table 5). Other studies have reported a similar finding showing that patients hospitalized with COVID-19 with a history of CVD and especially heart failure had increased mortality [5,18]. Krittanawong et al. found that patients with COVID-19 with pre-existing CVD were at the highest risk for myocardial injury and mortality among infected patients [16]. Goyal et al. attributed age and comorbidities to the increased risk of mortality in patients with COVID-19 that had a history of CVD [18]. The mortality rate we report associated with a history of CVD is somewhat lower than mortality rates reported early during the pandemic, especially in the first 3–6 months of the pandemic, of 15–30% [19,20,21]. A possible reason for the low mortality rate in this study could be the generally healthier immune profile reported for individuals in Japan compared to countries like the United States (US) [22]. Within the US, mortality or cardiovascular outcomes associated with COVID-19 were not found to differ by race [23].

We found that the most common CVDs in patients with a history of CVD were hypertension (33.7%), stroke (5.7%) and arrhythmias (5.1%) (Table 6). Several comorbidities have been reported to increase mortality in patients with COVID-19 including older age, especially age 70 and older, hypertension, and type II diabetes [24]. Around 30% of the patients in our study were >65 years of age (Table 1 and Figure 1) and 11.9% had type II diabetes and 3.3% obesity (Table 3). Similar to our study, the top comorbidities in a study of 5700 COVID-19 patients from New York area hospitals were hypertension (*n* = 3026; 56.6%), obesity (*n* = 1737; 41.7%), and diabetes (*n* = 1808; 33.8%) [19]. Thus, this study with a largely Japanese population agrees with earlier studies from other regions of the world that older age and comorbidities, especially cardiovascular comorbidities, contribute to mortality from COVID-19.

We found roughly 60% of the registry subjects were male and 40% female, consistent with other registries like the AHA COVID-19 CVD Registry and other large studies like the COVID-19 Expanded Access Program [4,19,21,25]. This male dominant sex ratio of 1.5:1 male to female occurred across cultures and ethnic groups/races with more men being hospitalized with COVID-19, being admitted to the ICU, and having a greater mortality than women [26,27,28,29]. However, the strongest predictor of death from COVID-19 is older age (70 years and older) [5]. In this study age of approximately 70 was associated with a history of CVDs, acute care complications and the need for respiratory support (Figure 1). Importantly, myocarditis occurs more often in males under the age of 50 and is more severe in males [7,8]. In this study, patients without a history of CVD were younger and had a lower mortality (Figure 1), and biomarkers from this demographic likely represented patients that could have myocarditis. It is not unusual for myocarditis to develop without distinct heart failure symptoms, which is one of the factors that can make myocarditis difficult to diagnose [30]. Thus, the mild heart failure symptoms observed in patients in this study (NYHA class I-II) do not indicate that patients did not have myocarditis. Imaging studies have found that myocardial inflammation often develops during COVID-19 and can persist after recovery from the infection in the absence of distinct cardiac heart failure symptoms [31,32]. However, when viral myocarditis is more severe, symptoms can present with heart failure, chest pain, and abnormal ECGs that can mimic an acute coronary syndrome, or ventricular arrhythmias [17,33] A number of studies have found that most patients with COVID-19 (around 90%) have alterations in ECG variables that are associated with increased mortality [34,35,36]. Acute myocardial injury with COVID-19 is characterized by elevated cardiac troponins accompanied by ST-segment elevation or depression on ECG; however, these findings are not sensitive in detecting myocarditis and their absence is not exclusionary [17,33]. These findings indicate COVID-19 patients should receive an ECG, cardiac troponin and other imaging tests such as cardiac MRI or echocardiography-derived global longitudinal strain to determine whether they could have myocarditis (and a biopsy if clinically warranted). It was difficult to determine the relationship of ECG or cardiac imaging in this study to myocarditis in COVID-19 cases because only around 7% of registry patients received an ECG and only 2% cardiac imaging (Table 6). Anti-coagulation therapy has also been found to be important in reducing mortality in COVID-19 patients [37]. However, coagulation is not a known complication related to myocarditis, which is mediated by an inflammatory response to the virus [8,9].

A small number of pregnant women were included in the study (*n* = 17, 2.5%). A number of studies have found that pregnant women with symptomatic SARS-CoV-2 infection are more likely to need intensive care, ventilation, and have a higher risk of pre-term deliver [38,39]. However, neonatal transmission is very rare, and several studies have found that there are no specific SARS-CoV-2 histopathologic placental changes observed in adverse perinatal outcomes, nor is there any evidence of a greater risk of spontaneous abortion, preeclampsia, pre-term delivery or stillbirth [40,41,42]. One large case–control study (1:2) (*n* = 71 COVID placentas vs. *n* = 142 control placentas) found no cases of the maternal–fetal transmission of SARS-CoV-2, and all the newborns were in good health at birth [43]. However, there were significant differences in decidual arteriopathy (40.9% vs. 1.4% *p* < 0.0001), decidual inflammation (32.4% vs. 0.7% *p* < 0.0001), perivillous fibrin deposition (36.6% vs. 3.5% *p* < 0.0001) and fetal vessel thrombi (22.5% vs. 0.7% *p* < 0.0001) [43], suggesting virus effects on the placenta without effects on the newborn. Future studies are needed to better understand whether these placental alterations influence the newborn. We did not specifically analyze cardiovascular outcomes in the pregnant cohort in this study.

Sera biomarkers that indicate cardiac damage, inflammation and thrombosis such as troponin T and I, CRP, tumor necrosis factor (TNF), IL-1β and IL-6 have been found to be associated with increased mortality as well as being elevated in males with COVID-19 compared to females [28,44]. Importantly, in this registry sera biomarkers were elevated in COVID-19 patients compared to reference levels regardless of CVD history. This suggests that cardiac damage and inflammation occurred in patients without a history of CVD, which may indicate undiagnosed myocarditis. Elevated troponin T and I levels indicate elevated cardiac damage or necrosis. Elevated NT-proBNP could indicate disturbances in myocardial ventricular systolic and diastolic function. Elevated CRP, IL-6 and Ig FLCs indicated increased inflammation. In the study by Saleh et al. from a total of 187 patients with confirmed COVID-19, almost 30% exhibited myocardial injury indicated by elevated troponin T [13]. Manke et al. found that elevated cardiac injury biomarkers were associated with generalized myocardial edema on cardiac MRI even though the echocardiogram was normal in patients with COVID-19 [31]. Overall, our findings suggest that mild myocardial injury and inflammation (myocarditis) could be caused by COVID-19 that goes unrecognized and thus not diagnosed. These findings are important because myocarditis is known to progress to DCM, especially in males [10,45]. Our findings suggest that physicians should carefully follow post-COVID patients, especially males, checking for the development of chronic cardiomyopathy or DCM. Our findings further indicate that even in cases where no abnormalities are found in ECG or ultrasound cardiography that myocardial damage may occur, and cardiovascular and inflammatory biomarkers may be useful for the diagnosis.

## 5. Limitations of the Study

Our REDCap survey was created in March 2020 at the beginning of the pandemic before the effect of SARS-CoV-2/COVID-19 was known on the heart. Thus, we did not include analysis of cardiovascular conditions such as myocardial infarct (MI) and postural orthostatic tachycardia syndrome (POTS) in the survey that are now known to be important outcomes of SARS-CoV-2/COVID-19 [46,47]. Myocarditis may not have been diagnosed by hospitals in this study due to the lack of access to cardiac MRI, which is the primary tool used to diagnose myocarditis aside from a cardiac biopsy. Most patients with COVID-19 (97.1%) in the registry were not assessed for CVD using echocardiogram or cardiac MRI. Because subjects did not receive a cardiac MRI or myocardial biopsy, conclusions regarding the relative contributions of demand ischemia, myocarditis and/or microvascular thrombosis to mortality cannot be drawn from the data. The registry also assessed only a relatively small number of patients. Additionally, the study did not examine patients that were exclusively from Japan. Results obtained from the registry in the first year of the pandemic reflect the SARS-CoV-2 variants that were present in Japan and Germany at that time and findings from later variants may vary.

## 6. Conclusions

We report cardiovascular complications from COVID-19 in the first year of the pandemic in a predominantly Japanese population. Mortality was increased by a history of CVD that included hypertension, stroke and arrythmias as well as other pre-existing conditions including type II diabetes, cancer, obesity, and kidney disease. Elevated sera cardiac troponins, IL-6, creatine kinase, D-dimer and Ig FLC biomarkers indicate cardiac damage and inflammation characteristic of myocarditis, especially in the COVID-19 patients without a history of CVD that are under 50 years of age. Few patients in this registry received an ECG (7%) or cardiac imaging test (2%), but an examination of serum biomarkers suggest that myocardial damage may have occurred from SARS-CoV-2/COVID-19, and that an examination of inflammatory biomarkers may be useful for the diagnosis.

## 7. Perspectives and Significance

Our study is novel in that it provides data on the response of the Japanese population to COVID-19 for which there are few published studies. This study confirms that during the first year of the pandemic these patients had similar risk factors and comorbidities as other regions of the world. Risk factors included age 70, a history of CVD, and male sex. We report for the first time a mortality rate around 10% for patients from Japan with COVID-19 that have a history of CVD. This rate is relatively low compared to other countries early in the pandemic. We also found that elevated sera biomarkers that indicate cardiac damage and inflammation are present in patients that are under 50 years of age and that did not have a history of CVD, which suggests that these patients could have undiagnosed myocarditis. The clinical implications of the study are that all patients with COVID-19 should receive an ECG and assessment of cardiac troponins to determine whether further cardiac evaluation should be undertaken, especially in patients with a history of hypertension or other cardiovascular risk factors.

## Figures and Tables

**Figure 1 diagnostics-12-02350-f001:**
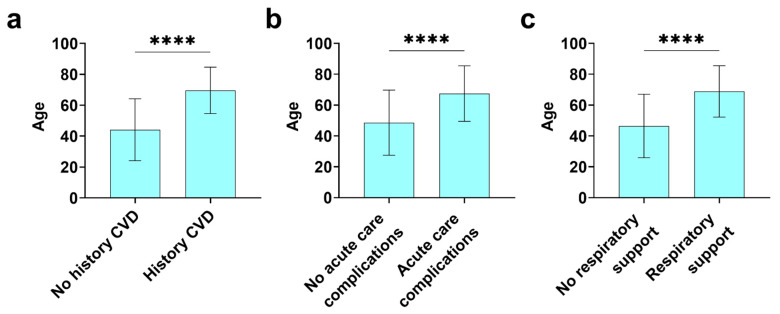
Patients with a history of CVD, acute care complications or in need of respiratory support are significantly older. Patient age at time of COVID infection was assessed between patients with (**a**) no history of CVD (*n* = 472) vs. a history of CVD (*n* = 207), (**b**) no acute care complications (*n* = 555) vs. acute care complications (*n* = 111), and (**c**) no respiratory support (*n* = 515) vs. respiratory support (*n* = 154). CVD history includes stroke, CAD (which included myocardial infarct), valvular heart disease, hypertension, arrhythmias, heart failure, myocarditis, cardiomyopathy, and prior CABG surgery. Acute care complications include pneumonia, ARDS, ICU admission, shock, AKI, and cardiac arrest. Respiratory support includes supplemental O_2_, NIPPV, mechanical ventilation/intubation, VV ECMO, and VA ECMO. Data shown as mean ± SD by Student’s *t* test; ****, *p* < 0.0001.

**Figure 2 diagnostics-12-02350-f002:**
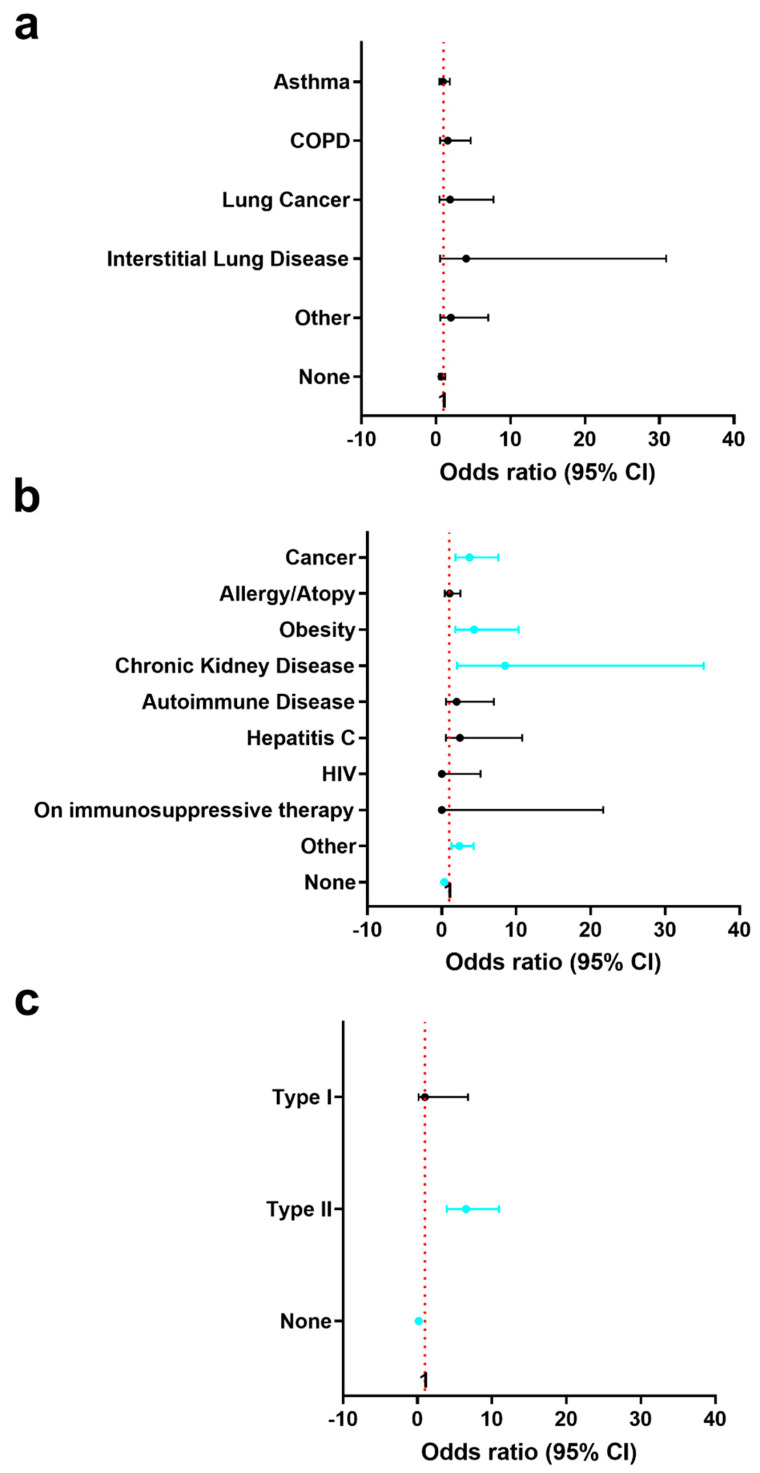
Patients with a history of CVD had significantly more cancer, obesity, chronic kidney disease and type II diabetes as pre-existing conditions. Patients with a history of CVD were compared to those without CVD to assess differences in (**a**) pre-existing lung disease history (*n* = 659), (**b**) other pre-existing condition history (*n* = 658) and (**c**) history of diabetes (*n* = 662). Data shown as mean ± 95% confidence interval by Fisher’s exact test (turquoise color: *p* < 0.05).

**Figure 3 diagnostics-12-02350-f003:**
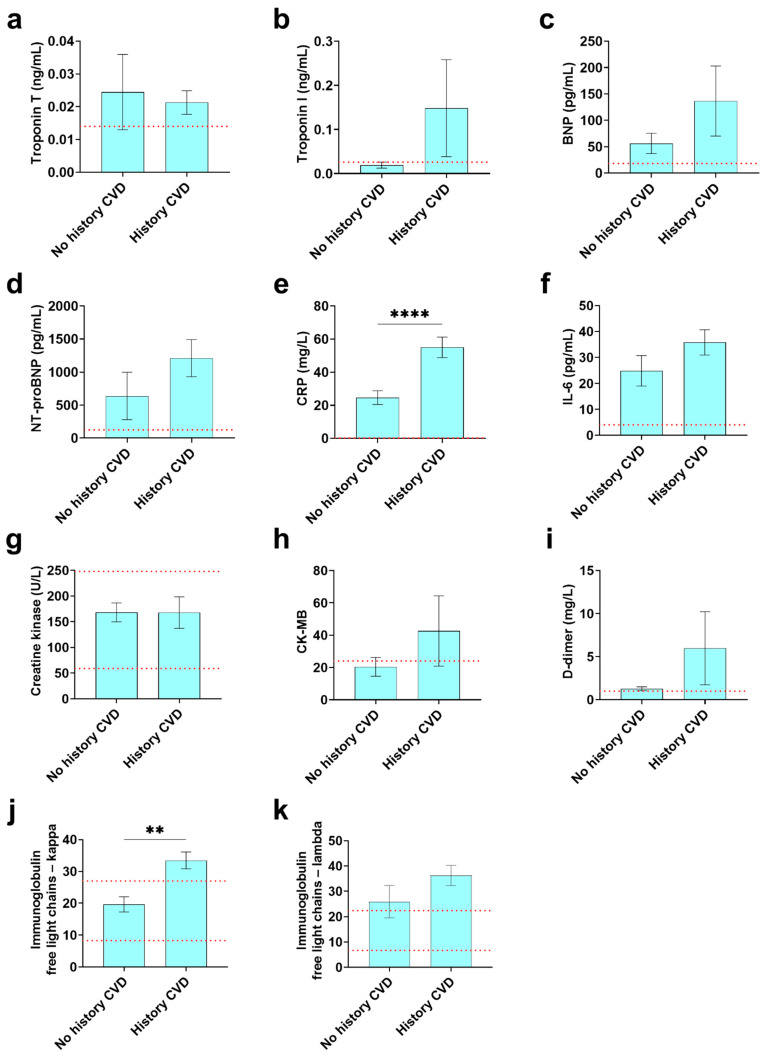
Patients regardless of history of CVD patients had elevated cardiac biomarkers. Cardiac and inflammatory biomarker levels were compared between patients with a history of CVD and those without CVD for (**a**) troponin T (*n* = 7/11), (**b**) troponin I (*n* = 9/17), (**c**) BNP (*n* = 13/8), (**d**) NT-proBNP (*n* = 29/47), (**e**) CRP (*n* = 184/103), (**f**) IL-6 (*n* = 84/37), (**g**) creatine kinase (*n* = 178/60) (**h**) CK-MB (*n* = 14/16), (**i**) D-dimer (*n* = 154/57), (**j**) kappa immunoglobulin FLC (*n* = 4/10), and (**k**) lambda immunoglobulin FLC (*n* = 6/10). Data shown as mean ± SEM by Student’s *t* test. Red dotted line: reference range for biomarker. (**, *p* < 0.01, ****, *p* < 0.0001).

**Figure 4 diagnostics-12-02350-f004:**
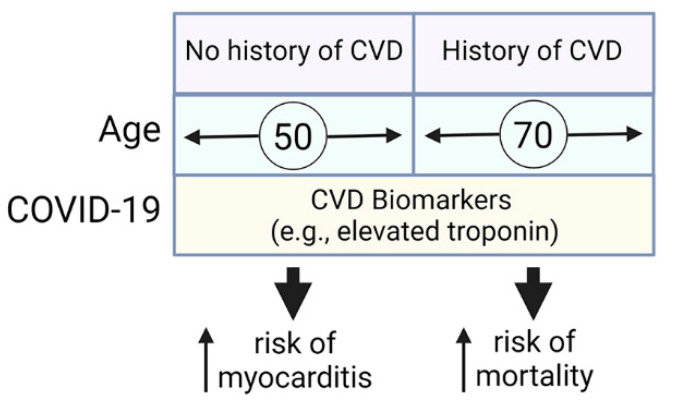
Summary of results. We compared patients with a history of cardiovascular disease (CVD) to those with no history and found that patients with a history of CVD were an average age of around 70 years old compared to patients without a history of CVD that were significantly younger, around age 50 years. Older patients with a history of CVD had an increased risk of mortality of around 10% compared to younger patients without a history of CVD at around 2%. COVID-19 led to an increase in many CVD biomarkers associated with myocardial damage and heart failure such as troponins and NT-proBNP. Patients with elevated CVD biomarkers from COVID-19 had increased risk of developing myocarditis in younger patients without a history of CVD and increased mortality in older patients with a history of CVD.

**Table 1 diagnostics-12-02350-t001:** Demographics of 696 patients in the COVID-19 Registry.

	Answered *n*	*n* (%)	Mean	SD
**Sex**	694	Male: 411 (59.2) Female: 283 (40.8)		
**Country**	695	Japan: 664 (95.5) Germany: 26 (3.7) Other: 5 (0.7)		
**Ethnicity**	688	Asian: 645 (93.8) Caucasian: 21 (3.1) Hispanic: 13 (1.9) Middle Eastern: 4 (0.6) Black: 2 (0.3) Pacific Islander: 0 (0.0) American Indian/Alaska Native/Indigenous: 0 (0.0) Other: 0 (0.0) Unknown: 3 (0.4)		
**Vulnerable Populations**	696	Elder (>65 years): 219 (31.5) Child (<18 years): 32 (4.6) Pregnant: 17 (2.4)		
**Pregnancy Trimester**	17	1st: 6 (35.3) 2nd: 2 (11.8) 3rd: 9 (52.9)		
**Age**	692	≤50 years: 331 (47.8) >50 years: 361 (52.2)	52.0	21.9

**Table 2 diagnostics-12-02350-t002:** COVID-19 symptoms and outcomes for patients in registry with or without history of CVD.

Answered *n* = 672 (467 No history CVD/205 History CVD) *^a^*	*n* (%)	No History CVD *n* (%)	History CVD *n* (%)	*p*-Value *^b^*
**Fever**	498 (74.1)	352 (75.4)	146 (71.2)	0.29
**Dry cough**	288 (42.9)	**214 (45.8)**	**74 (36.1)**	**0.022** * ^c^ *
**Fatigue**	252 (37.5)	173 (37.0)	79 (38.5)	0.73
**Loss of smell/taste**	177 (26.3)	**145 (31.0)**	**32 (15.6)**	**<0.0001**
**Shortness of Breath**	177 (26.3)	**110 (23.6)**	**67 (32.7)**	**0.017**
**Sore throat**	161 (24.0)	114 (24.4)	47 (22.9)	0.70
**Headache**	152 (22.6)	113 (24.2)	39 (19.0)	0.16
**Myalgia**	102 (15.2)	76 (16.3)	26 (12.7)	0.25
**Diarrhea**	98 (14.6)	75 (16.1)	23 (11.2)	0.12
**Decreased appetite**	92 (13.7)	**53 (11.3)**	**39 (19.0)**	**0.010**
**Chills**	57 (8.5)	34 (7.3)	23 (11.2)	0.10
**Chest discomfort**	33 (4.9)	25 (5.4)	8 (3.9)	0.56
**Nausea**	28 (4.2)	21 (4.5)	7 (3.4)	0.68
**Stomach/abdominal pain**	19 (2.8)	**9 (1.9)**	**10 (4.9)**	**0.043**
**Vomiting**	12 (1.8)	10 (2.1)	2 (1.0)	0.36
**Dizziness/Light-Headedness**	11 (1.6)	6 (1.3)	5 (2.4)	0.32
**Palpitations**	3 (0.4)	1 (0.2)	2 (1.0)	0.22
**Angina**	3 (0.4)	1 (0.2)	2 (1.0)	0.22
**Confusion/Altered mental status**	1 (0.1)	1 (0.2)	0 (0.0)	0.99
**Myocarditis**	0 (0.0)	0 (0.0)	0 (0.0)	0.99
**Other**	35 (5.2)	25 (5.4)	10 (4.9)	0.85
**None**	38 (5.7)	26 (5.6)	12 (5.9)	0.86

*^a^* CVD history group includes stroke, CAD (including prior MI), valvular heart disease, hypertension, arrhythmias, heart failure, myocarditis, cardiomyopathy, prior CABG. *^b^* Continuous data are expressed as mean ± standard deviation (SD) and *p*-values were obtained using Student’s *t* test. Categorical data are expressed as numbers and percentages and *p*-values were obtained using Fisher’s exact test. *^c^* Bold indicates factors that are significant comparing no history of CVD to history of CVD.

**Table 3 diagnostics-12-02350-t003:** History of pre-existing lung and other conditions in registry patients with or without a history of CVD.

	Answered *n* (no History CVD, History CVD)	Condition *n* (%)	No History CVD *n* (%)	History CVD *n* (%)	*p*-Value *^a^*
**History of Lung Conditions**	659 (466,193)	Asthma: 40 (6.1)	29 (6.2)	11 (5.7)	0.86
		COPD: 13 (2.0)	8 (1.7)	5 (2.6)	0.54
		Lung Cancer: 7 (1.1)	4 (0.9)	3 (1.6)	0.42
		Interstitial Lung Disease: 3 (0.5)	1 (0.2)	2 (1.0)	0.21
		Other: 9 (1.4)	5 (1.1)	4 (2.1)	0.46
		None: 589 (89.4)	421 (90.3)	168 (87.0)	0.21
**Other Pre-Existing Conditions**	658 (465,193)	**Cancer (excl. lung):** 34 (5.2)	14 (3.0)	20 (10.4)	**0.0003** * ^b^ *
		Allergy/Atopy: 23 (3.5)	16 (3.4)	7 (3.6)	>0.99
		**Obesity:** 22 (3.3)	8 (1.7)	14 (7.3)	**0.0011**
		**Chronic Kidney Disease:** 10 (1.5)	2 (0.4)	8 (4.1)	**0.0012**
		Autoimmune Disease: 9 (1.4)	5 (1.1)	4 (2.1)	0.46
		Hepatitis C: 6 (0.9)	3 (0.6)	3 (1.6)	0.37
		HIV: 2 (0.3)	2 (0.4)	0 (0.0)	0.99
		On immunosuppressive therapy: 1 (0.2)	1 (0.2)	0 (0.0)	0.99
		**Other:** 48 (7.3)	25 (5.4)	23 (11.9)	**0.0049**
		**None:** 533 (81.0)	401 (86.2)	132 (68.4)	**<0.0001**
**History of Diabetes**	662 (463,199)	Type I: 4 (0.6)	3 (0.6)	1 (0.5)	0.99
		**Type II:** 79 (11.9)	25 (5.4)	54 (27.1)	**<0.0001**
		**None:** 579 (87.5)	435 (94.0)	144 (72.4)	**<0.0001**

*^a^* Categorical data are expressed as numbers and percentages and *p*-values were obtained using Fisher’s exact test. *^b^* Bold indicates factors that are significant comparing no history of CVD to history of CVD.

**Table 4 diagnostics-12-02350-t004:** Lung imaging and respiratory support for registry patients with or without a history of CVD.

	Answered *n* (no History CVD, History CVD)	All Patients *n* (%)	No History CVD *n* (%)	History CVD *n* (%)	*p*-Value *^a^*
**Lung Imaging**	677 (471,206)	**Chest X-ray:** 277 (40.9)	203 (43.1)	74 (35.9)	0.089
		**CT:** 269 (39.7)	164 (34.8)	105 (51.0)	**0.0001** * ^b^ *
		**Both Chest X-ray and CT:** 171 (25.3)	117 (24.8)	54 (26.2)	0.70
		**No Imaging:** 302 (44.6)	221 (46.9)	81 (39.3)	0.078
**Lung Imaging Results**	369 (248,121)	**Ground glass shadowing:** 168 (45.5)	109 (44.0)	59 (48.8)	0.44
		**Pneumonia:** 90 (24.4)	48 (19.4)	42 (34.7)	**0.0018**
		**Bilateral patchy shadowing:** 60 (16.3)	36 (14.5)	24 (19.8)	0.23
		**Local patchy shadowing:** 14 (3.8)	10 (4.0)	4 (3.3)	0.99
		**Interstitial infiltrates:** 10 (2.7)	7 (2.8)	3 (2.5)	0.99
		**Pleural effusion:** 10 (2.7)	3 (1.2)	7 (5.8)	**0.017**
		**Interstitial abnormalities:** 6 (1.6)	4 (1.6)	2 (1.7)	0.99
		**CTR:** 1 (0.3)	1 (0.4)	0 (0.0)	0.99
		**Other:** 1 (0.3)	1 (0.4)	0 (0.0)	0.99
		**Normal:** 98 (26.6)	86 (34.7)	12 (9.9)	**<0.0001**
**Highest Level of Respiratory Support**	663 (465,198)	**Supplemental Oxygen:** 115 (17.3)	57 (12.3)	58 (29.3)	**<0.0001**
		**Mechanical Ventilation/Intubation:** 16 (2.4)	6 (1.3)	10 (5.1)	**0.0094**
		**NIPPV:** 13 (2.0)	6 (1.3)	7 (3.5)	0.069
		**VV Extra Corporeal Machine Oxygenation (ECMO):** 6 (0.9)	1 (0.2)	5 (2.5)	0.010
		**VA Extra Corporeal Machine Oxygenation (ECMO):** 0 (0.0)	0 (0.0)	0 (0.0)	0.99
		**None:** 513 (77.4)	395 (84.9)	118 (59.6)	<0.0001

*^a^* Categorical data are expressed as numbers and percentages and *p*-values were obtained using Fisher’s exact test. *^b^* Bold indicates factors that are significant comparing no history of CVD to history of CVD.

**Table 5 diagnostics-12-02350-t005:** Mortality and acute care complications in registry patients with or without a history of CVD.

	Answered *n* (no History CVD, History CVD)	All Patients *n* (%)	No History CVD *n* (%)	History CVD *n* (%)	*p*-Value *^a^*
**Mortality** * ^b^ *	677 (470,207)	29 (4.3)	8 (1.7)	21 (10.1)	**<0.0001** * ^c^ *
**Time in the ICU (Days)**	12 (5,7)	5.7 ± 3.5	5.8 ± 4.1	5.6 ± 3.3	0.92
**Time to death after admission (Days)**	26 (5,21)	18.0 ± 11.2	16.0 ± 17.3	18.4 ± 9.9	0.77
**Acute Care Complications**	663 (461,202)	**Pneumonia:** 91 (13.7)	38 (8.2)	53 (26.2)	**<0.0001**
		**Acute Respiratory Distress Syndrome (ARDS):** 16 (2.4)	5 (1.1)	11 (5.4)	**0.0016**
		**ICU Admission:** 13 (2.0)	5 (1.1)	8 (4.0)	**0.028**
		**Shock**: 6 (0.9)	1 (0.2)	5 (2.5)	**0.011**
		**Acute Kidney Injury (AKI):** 4 (0.6)	1 (0.2)	3 (1.5)	0.087
		**VT Cardiac Arrest:** 2 (0.3)	1 (0.2)	1 (0.5)	0.52
		**VF Cardiac Arrest:** 2 (0.3)	1 (0.2)	1 (0.5)	0.52
		**PEA Cardiac Arrest:** 2 (0.3)	0 (0.0)	2 (1.0)	0.093
		**Other:** 5 (0.8)	2 (0.4)	3 (1.5)	0.17
		**None:** 554 (83.6)	415 (90.0)	139 (68.8)	**<0.0001**

*^a^* Categorical data are expressed as numbers and percentages and *p*-values were obtained using Fisher’s exact test. ***^b^*** Mortality for index hospitalization. *^c^* Bold indicates factors that are significant comparing no history of CVD to history of CVD.

**Table 6 diagnostics-12-02350-t006:** Cardiovascular findings in registry patients with or without a history of CVD.

	Answered *n* (no History CVD, History CVD)	All Patients *n* (%)	No History CVD *n* (%)	History CVD *n* (%)	*p*-Value *^a^*
**CVD History**	681 (209,472)	**Hypertension** 159 (23.3)	0 (0)	159 (33.7)	**<0.0001** * ^b^ *
		**Stroke** 27 (4.0)	0 (0)	27 (5.7)	**<0.0001**
		**Arrhythmias** 24 (3.5)	0 (0)	24 (5.1)	**0.0002**
		**Coronary Artery Disease/MI** 15 (2.2)	0 (0)	15 (3.2)	**0.008**
		**Heart Failure** 9 (1.3)	0 (0)	9 (1.9)	0.06
		**Valvular Heart Disease** 5 (0.7)	0 (0)	5 (1.1)	0.33
		**Prior Coronary Artery Bypass Graft** 2 (0.3)	0 (0)	2 (0.4)	>0.99
		**Myocarditis** 1 (0.1)	0 (0)	1 (0.2)	>0.99
		**Cardiomyopathy** 1 (0.1)	0 (0)	1 (0.2)	>0.99
		**Other** 53 (7.8)	0 (0)	53 (11.2)	<0.0001
		**None** 209 (30.7)	209 (100.0)	0 (0)	<0.0001
**Cardiac Decompensation**	643 (459,184)	3 (0.5)	0 (0.0)	3 (1.6)	**0.023**
**NYHA Class**	614 (441,173)	**Class I** 32 (5.2)	17 (3.9)	15 (8.7)	**0.025**
		**Class II** 4 (0.7)	0 (0.0)	4 (2.3)	**0.0061**
		**Class I/II** 36 (5.9)	17 (3.9)	19 (11.0)	**0.0017**
		**Class III** 4 (0.7)	2 (0.5)	2 (1.2)	0.32
		**Class IV** 1 (0.2)	0 (0.0)	1 (0.6)	0.28
		**Class III/IV** 5 (0.8)	2 (0.5)	3 (1.7)	0.14
		**Unknown** 36 (5.9)	12 (2.7)	24 (13.9)	**<0.0001**
		**None** 537 (87.5)	410 (93.0)	127 (73.4)	**<0.0001**
**ECG Performed**	657 (462,195)	48 (7.3)	16 (3.5)	32 (16.4)	**<0.0001**
**ECG Abnormalities**	48 (16,32)	**Arrhythmia** 7 (14.6)	1 (6.3)	6 (18.8)	0.40
		**Atrial Fibrillation** 5 (10.4)	0 (0.0)	5 (15.6)	0.15
		**ST Elevation** 4 (8.3)	1 (6.3)	3 (9.4)	0.99
		**AV Block** 3 (6.3)	0 (0.0)	3 (9.4)	0.54
		**Non-sustained Ventricular Tachycardia** 2 (4.2)	1 (6.3)	1 (3.1)	0.99
		**QT Prolongation** 1 (2.1)	0 (0.0)	1 (3.1)	0.99
		**Abnormal Q** 0 (0.0)	0 (0.0)	0 (0.0)	0.99
		**ST Depression** 0 (0.0)	0 (0.0)	0 (0.0)	0.99
		**Atrial Flutter** 0 (0.0)	0 (0.0)	0 (0.0)	0.99
		**Other** 3 (6.3)	1 (6.3)	2 (6.3)	0.99
		**None** 26 (54.2)	13 (81.3)	13 (40.6)	**0.013**
**LVEF at admission**	8 (6,2)	60.4 ± 7.5	62.0 ± 8.1	55.5 ± 0.7	0.11
**Heart Imaging**	674 (469,205)	**Cardiac Echo** 12 (1.8)	6 (1.3)	6 (2.9)	0.20
		**CMR** 0 (0.0)	0 (0.0)	0 (0.0)	0.99
		**Cardiac Echo and CMR** 0 (0.0)	0 (0.0)	0 (0.0)	0.99
		**No Imaging** 662 (98.2)	463 (98.7)	199 (97.1)	0.20

*^a^* Categorical data are expressed as numbers and percentages and *p*-values were obtained using Fisher’s exact test. *^b^* Bold indicates factors that are significant comparing no history of CVD to history of CVD.

**Table 7 diagnostics-12-02350-t007:** Cardiovascular and inflammatory biomarkers found in COVID-19 registry patients with or without a history of CVD.

Biomarker	*n* (no History CVD, History CVD *^a^*)	No History CVD Mean ± SD *n* (%)	History CVD Mean ± SD *n* (%)	*Reference Range*	*p*-Value *^b^*
**Troponin T (ng/mL)**	18 (7,11)	0.02 ± 0.03 7 (38.9)	0.02 ± 0.01 11 (61.1)	*<0.014*	0.80
**Troponin I (ng/mL)**	26 (9,17)	0.02 ± 0.02 9 (34.6)	0.1 ± 0.5 17 (65.4)	*<0.026*	0.26
**BNP (pg/mL)**	21 (13,8)	56.2 ± 69.5 13 (61.9)	136.7 ± 187.7 8 (38.1)	*<18.4*	0.28
**NT-proBNP (pg/mL)**	76 (29,47)	637.7 ± 1940.5 29 (38.2)	1210.1 ± 1919.1 47 (61.8)	*<125*	0.21
**Peak CRP (mg/L)**	287 (184,103)	24.6 ± 56.8 184 (64.1)	55.0 ± 63.5 103 (35.9)	*<0.14*	**7.98 × 10^−5^** * ^c^ *
**IL-6 (pg/mL)**	121 (84,37)	24.8 ± 53.5 84 (69.4)	45.1 ± 73.8 37 (30.6)	*<4*	0.14
**Creatine kinase (U/L)**	238 (178,60)	169.4 ± 311.5 178 (74.8)	167.9 ± 237.5 60 (25.2)	*59–248*	0.97
**CK-MB**	30 (14,16)	20.4 ± 21.7 14 (46.7)	42.6 ± 87.0 16 (53.3)	*<24*	0.34
**D-dimer (mg/L)**	211 (154,57)	1.3 ± 2.8 154 (72.9)	6.0 ± 32.0 57 (27.0)	*<1*	0.27
**Immunoglobulin free light chains—kappa**	14 (4,10)	19.6 ± 4.8 4 (28.6)	33.5 ± 8.4 10 (71.4)	*8.3–27.0*	**0.0032**
**Immunoglobulin free light chains—lambda**	16 (6,10)	26.0 ± 15.7 6 (37.5)	36.3 ± 12.8 10 (62.5)	*6.7–22.4*	0.21

*^a^* CVD history group includes stroke, CAD (including prior MI), valvular heart disease, hypertension, arrhythmias, heart failure, myocarditis, cardiomyopathy, prior CABG. *^b^* Continuous data are expressed as mean ± standard deviation (SD) and *p*-values were obtained using Student’s *t* test. Categorical data are expressed as numbers and percentages and *p*-values were obtained using Fisher’s exact test. *^c^* Bold indicates factors that are significant comparing no history of CVD to history of CVD.

## Data Availability

The data presented in this study are available on request from the corresponding author. The data are not publicly available due to clinical patient data.

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
