# Peer review of "Cardiovascular Factors Associated with COVID-19 from an International Registry of Primarily Japanese Patients"

_diagnostics, 2022, doi:10.3390/diagnostics12102350_

Round 1
Reviewer 1 Report (Previous Reviewer 2)
I have no additional comment.
Author Response
Thank you for your review.
Reviewer 2 Report (New Reviewer)
I thank the Academic Editor and the Authors for allowing me to review this very interesting manuscript relating to cardiovascular manifestations in patients with SARS-CoV-2. I believe that this topic was very little debated at the beginning of the pandemic, and that studies like this can bring more knowledge on these types of complications. I don't feel like making any notes as I believe that both the introduction, the materials and methods (including statistical analysis), and the results are well written and presented, with very clear and exhaustive tables. The discussion was also well addressed and I suggest the authors to derive a paragraph that deals with the cardiovascular aspect in pregnant women in greater detail, perhaps referring more extensively to the aspect of Covid infection in this cohort of patients. In this regard, I suggest reading, studying, discussing and, therefore, citing the following papers.
Resta L, Vimercati A, Cazzato G, Mazzia G, Cicinelli E, Colagrande A, Fanelli M, Scarcella SV, Ceci O, Rossi R. SARS-CoV-2 and Placenta: New Insights and Perspectives. Viruses. 2021 Apr 21;13(5):723. doi: 10.3390/v13050723. PMID: 33919284; PMCID: PMC8143362.
Author Response
Thank you for bringing this to our attention. We have now written a paragraph in the discussion describing this important aspect of our findings.
This manuscript is a resubmission of an earlier submission. The following is a list of the peer review reports and author responses from that submission.
Round 1
Reviewer 1 Report
Matsumori and coworkers evaluated 696 patients in the COVID-19 Registry, 411 (59.2%) 32 were male and 283 (40.8%) were female. The average age was 52 years with 31.5% being >65 years of age. COVID-19 patients with a history of cardiovascular disease (CVD) had a greater percentage of the pre-existing conditions type II diabetes, cancer, obesity, and kidney disease. The authors concluded that even in cases where no abnormalities are found in ECG or ultrasound cardiography that myocardial damage may occur, and cardiovascular and inflammatory biomarkers may be useful for the diagnosis.
This study is timely and well-conducted. There are some issues that deserve clarifications:
11) The authors should include a subheading about the role of ecg in the prognosis. There are many articles that should be discuss (DOI: 10.1093/europace/euaa245; DOI: 10.1111/anec.12815; DOI: 10.1093/europace/euaa258). Moreover the authors must discuss the role of anticoagulant in this disease ( please cite DOI: 10.3389/fphar.2020.01124)
22) I suggest to insert a central figure to summarized the study results.
Reviewer 2 Report
Abstract. We developed an international registry to examine cardiovascular complications of COVID-19. A REDCap form was created in March 2020 at Mayo Clinic in collaboration with the International Society of Cardiomyopathy, Myocarditis and Heart Failure (ISCMF) and data was entered from April 2020 through April 2021. Of the 696 patients in the COVID-19 Registry, 411 (59.2%) were male and 283 (40.8%) were female with a sex ratio of 1.5:1 male to female. 95.5% of the patients were from Japan. The average age was 52 years with 31.5% being >65 years of age. COVID-19 patients with a history of cardiovascular disease (CVD) had a greater percentage of the pre-existing conditions type II diabetes (p<0.0001), cancer (p=0.0003), obesity (p=0.001), and kidney disease (p=0.001). They also had a greater mortality of 10.1% vs. 1.7% in those without a history of CVD (p<0.0001). The most common cardiovascular conditions in patients with a history of CVD were hypertension (33.7%), stroke (5.7%) and arrhythmias (5.1%) and one patient had myocarditis previously. However, no COVID-19 patients were diagnosed with myocarditis, although 6 pa-tients developed cardiac arrest. We found that troponin T, troponin I, brain natriuretic peptide (BNP), N-terminal pro-BNP (NT-proBNP), C-reactive protein (CRP), IL-6 and lambda immuno-globulin free light chains (Ig FLC) were elevated above reference levels in patients with COVID-19. Myocarditis is known to occur mainly in adults under the age of 50, and when we examined biomarkers in patients that were ≤50 years of age and had no history of CVD we found that a majority of patients on average had elevated levels of troponin T (71.4%), IL-6 (59.5%), creatine kinase/CK-MB (57.1%), D-dimer (57.8%), kappa Ig FLC (75.0%), and lambda Ig FLC (71.4%) suggestive of myocardial injury and possible myocarditis. We report the first findings to our knowledge of cardiovascular complications from COVID-19 in the first year of the pandemic in a predominantly Japanese population. Mortality was increased by a history of CVD and pre-existing conditions including type II diabetes, cancer, obesity, and kidney disease. Our findings indicate that even in cases where no abnormalities are found in ECG or ultrasound car-diography that myocardial damage may occur, and cardiovascular and inflammatory biomarkers may be useful for the diagnosis. The abstract is rumbling and difficult to read. Could you please divide it in different sections?
2) 1. Introduction L 56-59. Late in 2019 an outbreak of pneumonia with features of acute respiratory distress syndrome (ARDS) occurred in Wuhan, Hubei Province, China resulting in the World 58 Health Organization (WHO) naming the virus Severe Acute Respiratory Syndrome Coronavirus-2 (SARS-CoV-2) and the condition Coronavirus disease 2019 (COVID-19). Please improve this paragraph and add these references:
A- COVID-19 ARDS: getting ventilation right. Lancet. 2022 Jan 1;399(10319):22. doi: 10.1016/S0140-6736(21)02439-9.
B- Severe COVID-19 ARDS Treated by Bronchoalveolar Lavage with Diluted Exogenous Pulmonary Surfactant as Salvage Therapy: In Pursuit of the Holy Grail? J Clin Med. 2022 Jun 21;11(13):3577. doi: 10.3390/jcm11133577.
3) 1.Introduction. L82-85. The purpose of our registry was to examine cardiovascular complications of COVID-19 and to examine the occurrence of elevated heart failure biomarkers in patients without a prior history of CVDin order to assess the possible existence of cardiac damage or inflammation that may in- dicate myocarditis. Please improve this part and underline the clinical novelty of the study.
4) Tables. Demographics of 696 patients in the COVID-19 Registry. Please clear the decimal and approximate the percentage to a whole number. Could you please add r-value in all statistically significant parameters?
5) 4. Discussion L284-286 Relatively quickly in the pandemic it became clear that COVID-19 was associated with a number of cardiac complications including arrythmias, acute coronary syndromes, hypercoagulability, thrombosis, and myocarditis [14, 15]. Could you please summarise here the most important statistically significant results of the study?
6) 7. Perspectives and Significance L385-393. Little information is available on the response of the Japanese population to COVID- 19. This study confirms that during the first year of the pandemic that these patients had similar risk factors and comorbidities as other regions of the world. Risk factors included age 70, a history of CVD, and male sex. We report for the first time a mortality rate around 10% for patients from Japan with COVID-19 that have a history of CVD. This rate is relatively low compared to other countries early in the pandemic. We also found that elevated sera biomarkers that indicate cardiac damage and inflammation are present in patients that are under 50 years of age and that did not have a history of CVD, which suggests that these patients could have undiagnosed myocarditis. Please underline the novelty of the study and the clinical implicatios.